# An Overview of Recent Findings on Social Anxiety Disorder in Adolescents and Young Adults at Clinical High Risk for Psychosis

**DOI:** 10.3390/brainsci7100127

**Published:** 2017-10-11

**Authors:** Maria Pontillo, Silvia Guerrera, Ornella Santonastaso, Maria Cristina Tata, Roberto Averna, Stefano Vicari, Marco Armando

**Affiliations:** 1Child and Adolescence Neuropsychiatry Unit, Department of Neuroscience, Children Hospital Bambino Gesù, Piazza Sant’Onofrio 4, 00100 Rome, Italy; silvia.guerrera@opbg.net (S.G.); ornella.santonastaso@opbg.net (O.S.); mariacristinatata.mct@gmail.com (M.C.T.); roberto.averna@opbg.net (R.A.); stefano.vicari@opbg.net (S.V.); marco.armando@opbg.net (M.A.); 2Office Médico-Pédagogique Research Unit, Department of Psychiatry, University of Geneva School of Medicine, 1211 Geneva, Switzerland

**Keywords:** social anxiety disorder, Clinical High Risk, social functioning, psychotic symptoms

## Abstract

Background: Some studies have shown that anxiety is particularly frequent in the Clinical High Risk (CHR) for psychosis population. Notably, social anxiety disorder is identified as one of the most common anxiety disorders in CHR adolescents and young adults. Despite this, the frequency and the clinical significance of social anxiety in this population have been underestimated. Methods: A selective review of literature published between 2011 and 2017 on social anxiety disorder in CHR adolescents and young adults. Results: Five studies are included. In particular, three studies demonstrated that CHR adolescents and young adults have higher levels of anxiety compared to controls. Furthermore, anxiety, including social anxiety, is related to the severity of psychotic symptoms. The other studies included show inconsistent results regarding the possible relationship between social anxiety and social functioning. Conclusions: To date, the eidence concerning the comorbidity of social anxiety disorder and CHR in adolescents and young adults is not sufficient to provide clear guidelines for clinical practice. Future longitudinal studies on larger samples of the CHR adolescents and young adults are required to examine the relationship between social anxiety disorder and the presence of attenuated psychotic symptomatology.

## 1. Introduction

Recent years have seen an increase in clinical efforts towards intervention in the prodromal phase of the first psychotic episode. Criteria have been proposed to detect adolescents and young adults who are at Clinical High Risk (CHR) for psychosis. These criteria make a distinction between attenuated psychotic symptoms, brief intermittent psychotic symptoms, and genetic risk with a recent deterioration in functioning. Among the risk factors leading to a possible negative outcome for CHR adolescents and young adults, cannabis/use dependence and anxiety have an important role. Indeed, Serafini et al. [1] shows that cannabis use/dependence may be critically engaged in determining negative outcomes such as suicidal behavior among psychotic youths and CHR patients with those who attempt or complete suicide who showed some additional risk factors for suicide such as mood disorders, stressful life events, interpersonal problems, poor social support, lonely lives, and hopelessness.

Anxiety may be particularly common in the CHR population: several studies [2,3,4,5] have shown that 24–53% of CHR patients have an anxiety disorder. Moreover, anxiety disorder may be associated with severity of attenuated psychotic symptoms. Finally, in CHR patients, anxiety may be more stressful than sub-threshold psychotic symptoms [2,3,4,5].

Social anxiety disorder has been identified as one of the most frequent anxiety disorders diagnosed in CHR, at approximately 42% of individuals meeting the criteria for this disorder [6]. Despite this, there is a lack of evidence regarding the clinical significance of social anxiety disorder in the CHR population. Indeed, we have conducted a review to summarize and critically examine the evidence in the literature on the prevalence of social anxiety disorder in CHR population, and its relationship with the content and severity of attenuated psychotic symptoms (e.g., suspiciousness). Finally, we investigate the association between social anxiety disorder and functioning, especially in terms of social participation (social functioning).

## 2. Methods

### 2.1. Study Design

This paper consists of a selective review of the literature published between 2011 and 2017. The authors followed the Preferred Reporting Items for Systematic Reviews and Meta-Analyses (PRISMA) guidelines.

### 2.2. Search Strategy

A comprehensive literature search of the PubMed/MEDLINE, Cochrane Library, the Cumulative Index to Nursing and Allied Health Literature (CINHAL), PsycINFO, ClinicalTrial.gov databases was conducted. 

A search algorithm based on a combination of these terms was used: (social phobia or social anxiety disorder) and (clinical high risk psychosis or ultra high risk psychosis).

The last update of the search was on July 2017. 

### 2.3. Selection Criteria

The included studies investigated the prevalence and the clinical significance of social anxiety disorder in CHR adolescents and young adults. Therefore, we included only studies where the samples had an age range equal to 12–35 years with Clinical High Risk for psychosis and anxiety disorder, especially social anxiety disorder. We included only studies where Clinical High Risk for psychosis was assessed using a psychometrically validated scale (e.g., Structured Interview for Prodromal Syndromes and the Scale for Assessment of Prodromal Symptoms (SIPS/SOPS) [7] or Comprehensive Assessment of At Risk Mental State (CAARMS) [8]), and social anxiety disorder was diagnosed using recognized diagnostic criteria (e.g., International Classification of Disease 10th Revision (ICD-10) [9] or Diagnostic and Statistical Manual of Mental Disorders 5th (DSM-5) [10]).

Articles not in the area of interest of this review were excluded, e.g., review articles, conference proceedings, comments, editorials or letters. No language restrictions or study design restrictions were applied.

### 2.4. Selection Procedure, Data Extraction and Data Management

The bibliographies of the most relevant published articles in the area of our interest were examined.

Data on efficacy, acceptability and tolerability were extracted by four authors independently (M.C.T., O.S., S.G. and R.A.). Disagreements were sorted out in a consensus meeting by other reviewers (M.P., S.V. and M.A.). The search algorithm resulted in 110 articles, of which 40 referred to potentially eligible studies. Of these, 35 articles were non-empirical studies, reviews and commentaries.

In terms of evidence-based medicine, the quality of these studies was moderate.

Figure 1 presents a detailed flow diagram of the study selection process. 

## 3. Data Synthesis

We found a total of five studies on social anxiety disorder in CHR adolescents and young adults. Due to the low number of included studies, we proposed a narrative synthesis. The narrative synthesis required describing, organizing, exploring and interpreting the study findings, examining their methodological adequacy. 

## 4. Results

During the last seven years, five studies have been conducted, with results proving the presence of anxiety disorders, and especially social anxiety, in CHR adolescents and young adults. Details on the methodologies and results of the studies are shown in Table 1.

Recently McAusland et al. [5] investigated the diffusion of anxiety disorders in CHR adolescents and young adults, the clinical picture in the presence and absence of anxiety, the correlation between anxiety disorder at the initial stage and a possible transition to psychosis. The sample consisted of 765 CHR individuals matched with 280 healthy controls. CHR status was evaluated with the Structured Interview for Prodromal Syndromes [7] (SIPS/SOPS), mood and anxiety diagnosed with the Structured Clinical Interview for DSM-IV Disorders and severity of anxiety with the Social Interaction Anxiety Scale [15] (SIAS) and Self Rating Anxiety Scale [16] (SAS). The CHR group showed anxiety disorders more than the healthy control group; over 50% of the CHR sample presented conditions for anxiety disorders, in particular social phobia (*p* ≤ 0.001). 

The two anxiety rating scales, SIAS [15] and SAS [16], were significantly and positively correlated to symptoms on the SIPS/SOPS [7]; anxiety was related to attenuated psychotic and negative symptoms in CHR (*p* < 0.01). Furthermore, CHR with an anxiety disorder showed more suspiciousness (*p* < 0.001). No difference was found between CHR with anxiety disorders and CHR who had been evolving to psychosis (50.5%) than those who did not (50.9%) (*p* = 1.00).

Judith Rietdijk et al. [6] investigated the presence of anxious and depressive symptoms in a large CHR sample. This study tested the hypothesis that the severity of subclinical psychotic symptoms is correlated with the intensity of depressive and social anxiety symptoms and that female patients present more severe depressive and social anxiety symptomatology than male patients. Participants were 201 CHR. The results showed that 58% of CHR met the criteria for clinical depression as measured by the Beck Depression Inventory [17] (BDI) (*p* = 0.034), and 42% met the criteria for social phobia based on the SIAS [15] (*p* = 0.004). Both males and females had similar scores for intensity and frequency of positive symptoms measured by the Comprehensive Assessment of At Risk Mental States (CAARMS) [8]. Especially in female patients, anxiety occurred in association with the level of subclinical psychotic symptoms (*p* = 0.038) more than depression. Overall, female patients showed elevated levels of anxiety and depression compared to male patients. 

Wigman et al. [12] investigated the relationship between anxiety and depression disorders and the consequent onset of psychotic symptoms in a representative community sample of adolescents and young adults (*n* = 3021). The study was designed as a prospective longitudinal study, consisting of 4 data waves: the baseline (T0) and 3 follow-up waves at an average of, respectively, 1.6 (T0–T1, SD 0.2), 3.5 (T0–T2, SD 0.3) and 8.4 (T0–T3, SD 0.7) years after T0. The younger participants (14–17 years) were assessed 4 times, while subjects aged 18–24 years only 3 times. Psychotic symptoms measured by Munich-Composite International Diagnostic Interview [18] (CIDI) were reported in 27% of individuals with an anxiety disorder (e.g., panic disorder, generalized anxiety disorder, agoraphobia, specific phobias and social anxiety disorder, Post Traumatic Stress Disorder and Obsessive Compulsive Disorder or depressive symptoms in a large sample of adolescents and young adults from the general population. Individuals with anxiety and depression disorders were more likely to have psychotic symptoms than individuals without a disorder (*p* < 0.001).

Madsen et al. [13] investigated comorbidities, psychosocial difficulties and gender differences in a sample of 42 CHR in order to identify the needs of this population and to improve current treatment options. The sample was assessed with the CAARMS [8] and other psychiatric scales. In the CHR sample, the average functional level, measured with the Social and Occupational Functioning Assessment Scale [19] (SOFAS), was equal to 43.1 (“major impairment in several areas”). In the Global Functioning: Role Scale [20] (GF: Role) the average score was 4.6 (between “very serious” and “serious impairment independently”), and in the Global Functioning: Social Scale [21] (GF: Social) the average score of subjects was 5.8 (“moderate impairment in social functioning”). In association, a high comorbidity with mood and anxiety disorders was detected, peculiarly with social phobia and panic disorder, and the presence of a high risk of substance abuse, self-harming behavior and suicidal ideation, in particular in the onset phase.

Finally, Chudleigh et al. [14] investigated the correlation between social functioning and positive and negative symptoms, depressive disorder and social anxiety. The sample consisted of 20 patients with first episode psychosis (FEP) and 20 healthy controls. Symptoms of self-rated social anxiety measured by Brief Social Phobia Scale [22] (BSPS) appeared similar in patient groups, but patients with FEP and CHR (particularly those at risk) showed more severe anxiety than the controls.

Social anxiety symptoms were significantly related, with both qualitative and quantitative social functioning outcomes in the FEP group only. Remarkable symptoms of social anxiety were related to poorer independence-competence, and were connected to worse social functioning.

## 5. Discussion

The goal of this selective review was to contribute to the updating of recent findings about the prevalence and clinical significance of social anxiety disorder in adolescents and young adults at CHR for psychosis. 

In summary, all five studies included showed that the prevalence of anxiety disorders was significantly higher in CHR adolescents and young adults than in healthy controls. In large samples of CHR adolescents and young adults assessed by Rietdijk et al. [6] and McAusland et al. [5], social anxiety disorder was the most prevalent anxiety diagnosis (42% and 51% respectively). This finding is similar to Wigman et al. [12]. Interestingly, in these three studies, CHR participants with higher levels of anxiety also presented greater severity of attenuated psychotic symptoms. Particularly, in McAusland et al. [5], CHR participants with an associated anxiety disorder had higher level of suspiciousness than CHR participants without anxiety diagnosis. Social anxiety disorder may be the initial trigger of persecutory thinking. Alternatively, social anxiety disorder and suspiciousness may emerge together in early phases of psychosis, and evolve in a similar way. In addition, as argued by Micail et al. [23], social anxiety disorder may emerge in response to suspicious thoughts. 

As regards the association between social anxiety disorder and social functioning, in Madsen et al. [13], CHR adolescents and young adults had a low social functioning, codified as “severe impairment”. However, the absence of a control group and the small sample of CHR subjects included narrow the validity of this finding. Interestingly, in Chudleigh et al. [14], social anxiety symptoms were not dissimilar between CHR adolescents and young adults and FEP. In addition, social anxiety symptoms were significantly positively associated with impairment in social functioning only in the FEP group. Social anxiety may not yet be related to measures of social functioning in the CHR participants because of the early phase of the illness. 

Finally, of the five studies included, only McAusland et al. [5] investigated if anxiety disorders (especially social anxiety disorder) may be associated or not with the later transition to psychosis in CHR adolescents and young adults. This study found no differences in baseline anxiety severity and baseline anxiety disorder prevalence between CHR participants with transition to psychosis and CHR participants without transition to psychosis (follow-up: Two years). This study supported findings from three previous studies that found that anxiety did not predict transition to psychosis [24,25]. However, future longitudinal studies on larger samples of CHR adolescents and young adults are essential to examining the relationship between anxiety disorder and transition to psychosis.

Regarding the relationship between depression and attenuated psychotic symptoms, in our review, Wigman et al. [12] showed that individuals with depression disorders have been more likely to have psychotic symptoms than individuals without this disorder. Several studies with nonclinical adolescents and adults suggest that depression and affective dysregulation are also present at a subclinical level [26,27]. In addition, at a clinical level, depressive symptoms are present in the prodromal phases of at-risk individuals in the transition toward schizophrenia-spectrum disorders [28]. Overall, the presence of depressive symptoms seems to be a risk factor for psychosis in the longer term [27]. However, just as for anxiety, future longitudinal studies are essential to examine the relationship between depression and the transition to psychosis in CHR adolescents and young adults. Finally, none of the five studies included examines the possible relationship between substance use, social anxiety and attenuated psychotic symptoms in CHR adolescents and young adults. Indeed, Russo et al. [29] shows that social anxiety is relatively frequent (13%) in adolescent and young adult drug-users with clinical high risk of psychosis. Further study would be needed to clarify this. 

## 6. Implications for Practice and Research

Social anxiety disorder was common in CHR adolescents and young adults, and correlated with more severe attenuated psychotic symptoms (e.g., suspiciousness). Thus, assessment of social anxiety disorder, along with other anxiety disorders, should be standard for all CHR adolescents and young adults presenting for help. Also, treatment of social anxiety disorder and other anxiety disorders would be a goal for CHR adolescents and young adults, for instance through cognitive behavioral therapy (CBT). Indeed, CHR adolescents and young adults may be supported with a CBT approach, in order to enhance critical insight and recognition of cognitive biases regarding their content.

## 7. Limitations

Some limitations should be considered in our review. Firstly, there is a discordance of study design and measures for clinical assessment between the included studies. In addition, they examined mixed samples of CHR and FEP, and considered different outcomes. This does not allow a quantitative analysis of the results. Secondly, no studies investigated social anxiety disorder in a selective and specific way, but rather had a focus on anxiety disorder in general. This limits the interpretability of the findings. 

## 8. Conclusions

This selective review provided a synthesis of the recent literature on the prevalence and clinical significance of social anxiety disorder in CHR adolescents and young adults. Although it may not play a role in later transition to psychosis, anxiety—especially social anxiety—was common in this large sample of young people at CHR for psychosis. Future studies are needed. In particular, longitudinal future studies should be conducted, collecting information on the onset and clinical course of social anxiety disorder. This would make it possible to define whether social anxiety preceded or followed the onset of CHR criteria, and its clinical course between baseline and the transition to frank psychosis.

## Figures and Tables

**Figure 1 brainsci-07-00127-f001:**
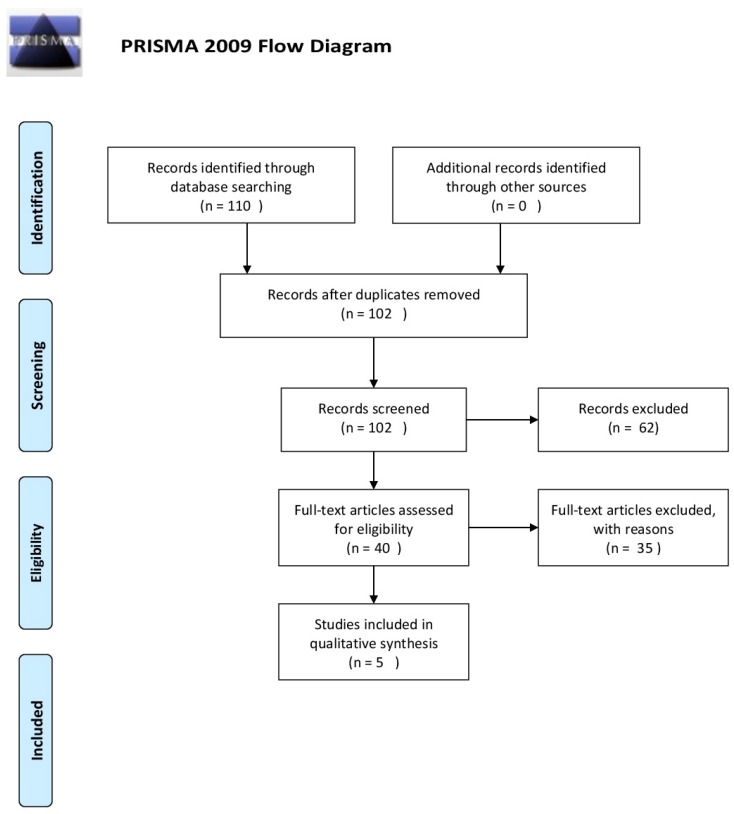
PRISMA flow diagram [11].

**Table 1 brainsci-07-00127-t001:** Results of included studies.

Study	Sample	Methods	Criteria for Diagnosis	Measure	Results
Mc Ausland et al. (2017) [5]	*n* = 765 CHR	Multicentric experimental study	Clinical High Risk	SIPS-SOPS	Attenuated Psychotic Symptom scores were correlated with higher SIAS (r_S_ = 0.12, *p* < 0.01) and SAS scores (r_S_ = 0.18, *p* < 0.001)
	280 Controls				
	Range age: 12–35			SAS	
					Severe negative symptoms were associated with higher SIAS (r_S_ = 0.33, *p* < 0.001) and SAS score (r_S_ = 0.18, *p* < 0.001)
	IQ > 70			SIAS	
Rietdijk et al. (2011) [6]	*n* = 201	Experimental	Clinical High Risk	PQ	Social Anxiety (SIAS > 36):42% CHR
				CAARMS	
	Mean age: 22.7			BDI-II	Gender and Social Anxiety:
			Personal Assessment and Crisis Evaluation (PACE) criteria		35.4% male vs. 49% female (*p* = 0.05)
	SD = 5.52				
				CDS	In female patients anxiety predicted Positive Symptom Score (*p* = 0.038)
				SIAS	
				Demographic questionnaire	
Wigman et al. (2012) [12]	*n* = 3021 (T0)	Longitudinal study	Disorders of Anxiety/Depression	(DIA-X/M-CIDI)	27% of individuals with disorders of anxiety/depression reported psychotic symptoms at any time point (36% at T2; 19% at T3) vs. 14% in those without (*p* < 0.001).
				SCL-90R	
				M-CIDI:	
	*n* = 2548 (T2)			-Caseness:	
				-Help seeking Behaviour;	
	*n* = 2210 (T3)			-Substance Use;	
				-Trauma;	
	Range age: 14–24			-Recent Life Events;	
				-Urbanicity	
				-Familial History of Help Seeking	
Madsen et al. (2017) [13]	*n* = 42	Experimental study	Children High Risk	CAARMS SCID-I	Social and Occupational Functioning Assessment Scale (SOFAS): Mean 43.1 (SD6.4), (*p* = 0.87).
	Range age: 18–40			SCID-II	
	Mean age = 23.8			SOFAS	GF: Role: Mean 4.6 (SD 1.12), (*p* = 0.97).
	SD = 4.7			YMRS	
				YMRS BPRS	GF: Social: Mean 43.1 (SD 5.8), (*p* = 0.045).
				SANS	
				MADRS	
				ASSIST	CHR presented major depressive disorders (79%), anxiety (74%), alcohol (31%) and cannabis (24%).
				GF: Role	
				GF: Social	
Chudleigh et al. (2011) [14]	20 FEP	Experimental study	CHR	CAARMS	FEP and CHR participants significantly different from control on quantitative (*p* = 0.000) and qualitative (*p* < 0.001) social functioning (SFS).
	20 CHR			BPRS	
	20 controls				
				SFS	
			FEP	SOFAS	Only in the FEP group (*p* < 0.01), social anxiety symptoms were associated with both qualitative and quantitative social functioning outcomes
	Range Age: 16–30			WHODAS	
				DASS	

**Abbreviations and Explanations**: PQ: Prodromal Questionnaire; CAARMS: Comprehensive Assessment of At Risk Mental State; BDI-II: Beck Depression Inventory-II; CDS: Calgary Depression Scale; SIAS: Social Interaction Anxiety Scale; RCT: Randomized Clinical Trial; SCL-90R: Symptom Checklist-90-R ;DIA-X/M-CIDI: Munich-Composite International Diagnostic Interview; SOFAS: Social and Occupational Functioning Assessment Scale; SCID I: Structured Clinical Interview for DSM-IV Axis I disorders; SCID II: Structured Clinical Interview for DSM-IV Axis II disorders; SANS: Scale for the Assessment of Negative Symptoms; MADRS: Montgomery-Asberg Depression Rating Scale; SIPS-SOPS: Structured Interview for Prodromal Syndromes and the Scale for Assessment of Prodromal Symptoms; YMRS: Young Mania Rating Scale; BPRS-E Brief Psychiatric Rating Scale-Expanded version; ASSIST: Alcohol, Smoking and Substance Involvement Screening Test; GF:Role: Global Functioning: Role Scale; GF: Social: Global Functioning: Social Scale; FHI: Family History Index; SAS: Zung Self Rating Anxiety Scale; SFS: Social Functioning Scale; WHODAS: World Health Organization Disability Assessment Scale II; DASS: Depression Anxiety Stress Scale.

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
