# Peer review of "An Overview of Recent Findings on Social Anxiety Disorder in Adolescents and Young Adults at Clinical High Risk for Psychosis"

_brainsci, 2017, doi:10.3390/brainsci7100127_

Round 1

Reviewer 1 Report

This is, in summary, a selective and detailed review of the current literature published between 2011 and 2017 on social anxiety disorder in CHR adolescents and young adults. The authors found, according to the five included studies, that CHR adolescents and young adults showed higher levels of anxiety compared to controls. Moreover, social anxiety resulted related to the severity of psychotic symptoms. Based on the other included studies, the authors concluded about the inconsistency of results regarding the possible relation between social anxiety and social functioning.

The paper is interesting and well-written in its current version, thus, in my opinion, only minor changes are needed. The authors may find as follows my main comments/suggestions.

First, when the authors correctly cited throughout the Introduction section of their paper the criteria that have been proposed to detect adolescents and young adults who are at Clinical High Risk (CHR) for psychosis, they may, at least briefly, focus on important risk factors that are documented in determining negative outcomes in CHR and first-episode psychosis subjects such as suicidal behavior. For instance, cannabis use/dependence has been reported to enhance suicide risk in adolescence, particularly when this use/dependence is associated with CHR and psychosis. Cannabis use/dependence may be critically involved in determining negative outcomes such as suicidal behavior among psychotic youths with those attempting or completed suicide who reported some additional risk factors for suicide such as mood disorders, stressful life events, interpersonal problems, poor social support, lonely lives, and feelings of hopelessness. In order to focus on this issue, i suggest to cite and discuss the review paper of Serafini and colleagues which has been published on Curr Pharmacol Des in 2012.

Furthermore, the fact that most of the included studies investigated a mixture of CHR and FEP patients and that samples included different measurements and different outcomes could be included as further additional limitations throughout the main text.

Finally, the manuscript might be revised by a native English speaker for the quality of Language.

Author Response

Response to Reviewers' Comments:

We thank the reviewers for their scrutiny of the manuscript and insightful remarks, their good feedback on our study was very encouraging. We hope to match their thoroughness and detail in our reply. Please, note that our replies are written in italics.

Reviewer #1:

The paper is interesting and well-written in its current version, thus, in my opinion, only minor changes are needed. The authors may find as follows my main comments/suggestions. 

1. First, when the authors correctly cited throughout the Introduction section of their paper the criteria that have been proposed to detect adolescents and young adults who are at Clinical High Risk (CHR) for psychosis, they may, at least briefly, focus on important risk factors that are documented in determining negative outcomes in CHR and first-episode psychosis subjects such as suicidal behavior. For instance, cannabis use/dependence has been reported to enhance suicide risk in adolescence, particularly when this use/dependence is associated with CHR and psychosis. Cannabis use/dependence may be critically involved in determining negative outcomes such as suicidal behavior among psychotic youths with those attempting or completed suicide who reported some additional risk factors for suicide such as mood disorders, stressful life events, interpersonal problems, poor social support, lonely lives, and feelings of hopelessness. In order to focus on this issue, I suggest to cite and discuss the review paper of Serafini and colleagues which has been published on Curr Pharmacol Des in 2012.

Response:

We thank the reviewer for the comment. Introduction has been amended as follows:

Among the risk factors leading a possible negative outcome for CHR adolescents and young adults, cannabis/use dependence and anxiety have an important role. Indeed, Serafini et al. [1] shows that cannabis use/dependence may be critically engaged in determining negative outcomes such as suicidal behavior among psychotic youths and CHR patients with those attempting or completed suicide who showed some additional risk factors for suicide such as mood disorders, stressful life events, interpersonal problems, poor social support, lonely lives, hopelessness.

2. The fact that most of the included studies investigated a mixture of CHR and FEP

patients and that samples included different measurements and different outcomes could be included as further additional limitations throughout the main text. 

Response: 

We agree with the reviewer. We have amended the text (section Limitations) according to this suggestion as follows:

  Firstly, there is a discordance of study design and measures for clinical assessment between included studies. In addition, they examined samples mixed of CHR and FEP and considered different outcomes. This does not allow a quantitative analysis of the results. Secondly, all studies did not investigate social anxiety disorder in a selective and specific way, but they had a focus anxiety disorder in general. This limits the interpretability of the findings.

3. The manuscript might be revised by a native English speaker for the quality of Language.

Response: Thank you for your suggestion.

The language of the manuscript was revised by Marco Armando. Marco Armando is native speaker of English.

Response to Reviewers' Comments:

We thank the reviewers for their scrutiny of the manuscript and insightful remarks, their good feedback on our study was very encouraging. We hope to match their thoroughness and detail in our reply. Please, note that our replies are written in italics.

Reviewer #1:

The paper is interesting and well-written in its current version, thus, in my opinion, only minor changes are needed. The authors may find as follows my main comments/suggestions. 

1. First, when the authors correctly cited throughout the Introduction section of their paper the criteria that have been proposed to detect adolescents and young adults who are at Clinical High Risk (CHR) for psychosis, they may, at least briefly, focus on important risk factors that are documented in determining negative outcomes in CHR and first-episode psychosis subjects such as suicidal behavior. For instance, cannabis use/dependence has been reported to enhance suicide risk in adolescence, particularly when this use/dependence is associated with CHR and psychosis. Cannabis use/dependence may be critically involved in determining negative outcomes such as suicidal behavior among psychotic youths with those attempting or completed suicide who reported some additional risk factors for suicide such as mood disorders, stressful life events, interpersonal problems, poor social support, lonely lives, and feelings of hopelessness. In order to focus on this issue, I suggest to cite and discuss the review paper of Serafini and colleagues which has been published on Curr Pharmacol Des in 2012.

Response:

We thank the reviewer for the comment. Introduction has been amended as follows:

Among the risk factors leading a possible negative outcome for CHR adolescents and young adults, cannabis/use dependence and anxiety have an important role. Indeed, Serafini et al. [1] shows that cannabis use/dependence may be critically engaged in determining negative outcomes such as suicidal behavior among psychotic youths and CHR patients with those attempting or completed suicide who showed some additional risk factors for suicide such as mood disorders, stressful life events, interpersonal problems, poor social support, lonely lives, hopelessness.

2. The fact that most of the included studies investigated a mixture of CHR and FEP

patients and that samples included different measurements and different outcomes could be included as further additional limitations throughout the main text. 

Response: 

We agree with the reviewer. We have amended the text (section Limitations) according to this suggestion as follows:

  Firstly, there is a discordance of study design and measures for clinical assessment between included studies. In addition, they examined samples mixed of CHR and FEP and considered different outcomes. This does not allow a quantitative analysis of the results. Secondly, all studies did not investigate social anxiety disorder in a selective and specific way, but they had a focus anxiety disorder in general. This limits the interpretability of the findings.

3. The manuscript might be revised by a native English speaker for the quality of Language.

Response: Thank you for your suggestion.

The language of the manuscript was revised by Marco Armando. Marco Armando is native speaker of English.

Reviewer 2 Report

This is an interesting review of the literature, addressing a still controversial topic: the role of anxiety (particularly social phobia) in youngsters at clinical high risk for psychosis.

The manuscript is understandable to a wide audience since it is clearly written. The search method is rigorous, the results are expressed in a comprehensive way, the graphic material is adequate.

The authors state that the role of anxiety in terms of cause or consequence is still debated, but I suggest deepening the subject.

For example, it should be interesting to discuss the possible link between social anxiety and the so-called delusional mood (the fearful perception of change that can precede frank psychosis). In addition, since in the reported studies depressive symptoms were recorded, it could be useful to discuss the association between depression and attenuated psychotic features (see for example Pedrero's article "Schizotypal traits and depressive symptoms in nonclinical adolescents" and Moritz's "Do depressive symptoms predict paranoia or vice versa?"). Moreover, I suggest to clearly state if some particular correlation has been found in terms of substance use, social anxiety and high risk of psychosis in the selected studied  (in fact, social phobia is relatively frequent in drug-users dysplaying high risk of psychosis. See Russo's "Substance use in people at clinical high-risk for psychosis).

Author Response

In the file word attached you can find my response

Round 2

Reviewer 2 Report

The authors have improved their paper following the reviewers' suggestions.

I do not have other comments for this manuscript.